# Water Pollution and Pollution–Control Capacity in Chinese Provinces: Panel Estimations of Provincial Environmental Kuznets Curves

Hiroyuki Taguchi [1,]*, Takeshi Fujino [2], Hidekatsu Asada [1] and Jun-Jun Ma [2]

[1]   Graduate School of Humanities and Social Sciences, Saitama University, Saitama 338-8570, Japan
[2]   Graduate School of Science and Engineering, Saitama University, Saitama 338-8570, Japan
*   Correspondence: htaguchi@mail.saitama-u.ac.jp

**Abstract:** China's rapid industrialization and urbanization in recent decades have deteriorated its water resource quality. This study focused on water pollution in terms of the chemical oxygen demand (COD) and the levels of ammonia nitrogen in the industrial and household discharges of different Chinese provinces. Although the heterogeneity of environmental Kuznets curves (EKCs) from Chinese provinces has been studied, the positions of provincial EKCs (which reflect the province-specific pollution effects not affected by the provincial income levels) have not been investigated to date. Therefore, through a factor analysis of the heterogeneity of provincial pollution under the EKC framework, we investigated how the capacity shortage for pollution control contributes to the provincial pollution levels. We also evaluated the heterogeneity of the EKCs from the provinces in terms of their positions (not their shapes), using a fixed-effect model to extract the province-specific pollution effects. The capacity shortage for pollution control, as one of the pollution factors, accounted for (a) 30% of industrial COD and industrial ammonia nitrogen; (b) 60% of household COD; and (c) 80% of household ammonia nitrogen. Our results indicate that China still has a large capacity to mitigate water pollution via policies and by building the capacity for pollution control through the development and training of human resources.

**Keywords:** environmental Kuznets curves; chemical oxygen demand; ammonia nitrogen; Chinese water issues

## 1. Introduction

China's growth has significantly improved the country's living standards since the implementation of the Open-door policy and the Reform Policy in 1978. The economic status of China was promoted from the low-income category to the lower-middle-income category in 1997, and to upper-middle-income category in 2010, based on the World Bank income classification. However, this rapid economic development resulted in serious damages to its environment through industrialization and urbanization. Water pollution is one of the vital issues that influences the survival of human beings and the development of socio-economic systems. According to the Environmental Performance Index, China remains in 80th place among 180 countries in terms of water resources. To address the issue of water pollution, the Chinese government has set numerical targets to reduce the two main water pollutants: chemical oxygen demand (COD) discharge, after the 11th Five-Year Plan (2006–2010), and ammonia nitrogen discharge, after the 12th Five-Year Plan (2011–2015). The current 14th Five-Year Plan (2021–2025) contains binding targets to reduce COD and ammonia nitrogen discharges by 8% during the planned period. These targets have been almost achieved through policy efforts; however, the pollution discharges still remain massive, keeping the water quality at a low level. The groundwater supplies in more than half of the Chinese cities were categorized as "bad to very bad," while more than a quarter of China's major rivers were considered "unfit for human contact" in 2014 [1,2].

In addition to the nation-wide issue of water pollution, another vital concern in China is the regional heterogeneity of the pollution levels and the factors influencing them. The water pollution levels largely differ by provinces, and the industrialization, urbanization and pollution-control capacities that affect pollution levels vary by provinces. There are also policy priority areas for water pollution control that have been set by the Chinese government [3]. (COD and ammonia nitrogen originate from the same sources, and the major industries that emit these pollutants are papermaking, textile and chemical ones (Cai et al. [1], Liu et al. [4]). China's sewage treatment mainly adopts six major sewage treatment processes: the oxidation ditch process, the A2/O process, the traditional active sewage treatment method, the SBR process, the A/O process, and the biofilm process (Ministry of Ecology and Environment of China [5]).

This study focused on the water pollution measures that target COD and ammonia nitrogen from industrial and household discharges in Chinese provinces, and aimed to investigate the contribution of capacity shortage to the pollution control of provincial pollution levels, through a factor analysis that evaluated the heterogeneity of provincial pollution under the analytical framework of the environmental Kuznets curve (EKC). We performed the following steps: (1) the EKC was estimated econometrically from the provincial panel data using a fixed-effect model; (2) the province-specific pollution effect was extracted from the fixed effect, which is not affected by the provincial income level, on the EKC; (3) the alternative EKC was re-estimated by replacing the fixed-effect model with the possible contributors to the province-specific pollution, such as the capacity for pollution control, the industrialization degree (for industrial discharges), and the urbanization degree (for household discharges); and (4) the contribution of the capacity shortage to the pollution control of the province-specific pollution level was quantified through a factor analysis.

The main finding of this study is that the capacity shortage for pollution control, as one of the pollution factors, accounts for (a) about 30% of industrial COD and industrial ammonia nitrogen; (b) about 60% of household COD; and (c) about 80% of household ammonia nitrogen. Therefore, China still has much capacity to mitigate COD-related species and ammonia nitrogen via policy change.

The remainder of the paper is structured as follows. Section 2 reviews the literature related to the EKC issues, including water pollution in China, and clarifies this study's contributions. Section 3 shows the materials and methods for the empirical study. Section 4 presents the estimation results and the discussion. Section 5 summarizes and concludes this paper.

## 2. Literature Review and Contributions

The EKC provides an analytical framework to examine how economies deal with environmental issues. It postulates an inverted-U-shaped relationship between pollution and economic development. Kuznets's name was apparently attached to the curve by Grossman and Krueger [6], who noted its resemblance to Kuznets' inverted-U relationship between income inequality and development. The EKC dynamic process is defined as follows. In the first stage of industrialization, pollution worsens rapidly because people are more interested in jobs and income than in clean air and water, and environmental regulation is correspondingly weak. Along the curve, pollution reduces in wealthy societies, because leading industrial sectors become cleaner, people value the environment, and regulatory institutions become more effective [7].

Since the report of the World Bank [8] initially discussed EKC issues, empirical tests and theoretical debates have intensified, supporting the applicability of EKC for some regions and environment problems [9–12]. At the initial stage until the 1990s, most of the empirical studies focused on validating the EKC hypothesis and its requirements using cross-sectional data. Since the late 1990s, however, the EKC studies have shifted from cross-sectional analyses to time-series analyses, and more importantly, have examined the heterogeneity of EKCs from individual economies, in terms of the curve's shapes and positions. In this context, Dasgupta et al. [7] presented three different EKC scenarios from the

conventional inverted-U EKC: Race to the Bottom (pessimistic, with a continuation of the highest level of pollution), New Toxics (pessimistic with a higher curve, owing to the newly emerging pollutants), and Revised EKC (optimistic with a lower and flatter curve, owing to a better management of pollution). These scenarios have been subjected to empirical tests [13–16]. Sarkodie and Strezov [17] comprehensively reviewed the heterogeneity of the EKC modalities in terms of the curve's shapes and positions.

There is a large body of literature on EKC studies for several countries and for several levels of environmental quality; however, studies on the EKC of China have increased since the 2000s. Therefore, there is a relatively limited number of EKC studies, particularly on water pollution in China, that cover total provinces or specific areas (Table 1). Their estimations show ambiguous and mixed outcomes; some studies identify the validity of the inverted-U-shaped EKC [2,18–21], whereas the others demonstrate that the EKC modality is dependent on regions and pollutants [1,4,22–24].

**Table 1.** Literature review of environmental Kuznets curve studies on water pollution in China.

|  | Sample Areas | Pollutants | Summary |
| --- | --- | --- | --- |
| Cai et al. (2020) [1] | 31 provinces | WW, COD, $NH_4$-N | Modality of EKC depends on regions |
| Liu et al. (2019) [22] | Shandong | WW, COD, $NH_3$-N | Modality of EKC depends on pollutants |
| Zhang et al. (2017) [2] | 27 provinces | COD, $NH_3$-N | Inverted-U shaped EKC is identified |
| Zhao et al. (2017) [18] | 31 provinces | water use | Inverted-U shaped EKC is identified |
| Wang et al. (2017) [23] | Urumqi | WW, COD, $NH_3$-N | Modality of EKC depends on pollutants |
| Li et al. (2016) [19] | 28 provinces | WW | Inverted-U shaped EKC is identified |
| Liu et al. (2016) [4] | Zaozhuang | WW, COD, $NH_3$-N | Modality of EKC depends on pollutants |
| Jayanthakumaran & Liu (2012) [20] | 31 provinces | COD | Inverted-U shaped EKC is identified |
| Liu et al. (2007) [24] | Shenzhen | TPH, etc. | Modality of EKC depends on pollutants |
| Shen (2006) [21] | 31 provinces | COD, Arsenic, Cadmium | Inverted-U shaped EKC is identified |

Notes: WW: waste water discharge; COD: chemical oxygen demand; $NH_3$-N and $NH_4$-N: ammonia nitrogen; TPH: total petroleum hydrocarbon; Sources: Authors' description.

With respect to the heterogeneity of EKCs from Chinese provinces, Cai et al. [1] demonstrated the varying types of EKCs depending on the province: "good EKCs" (negative monotonic shape, inverted N-shape, inverted U-shape, and M-shape), "bad EKCs" (positive monotonic shape, N-shape, and U-shape), and "transition EKCs" (positive monotonic and flat-tailed shape). To the best of our knowledge, there are no studies investigating the "positions" of provincial EKCs and that reflect the province-specific pollution effects that are not affected by the provincial income levels. Therefore, this study focused on analyzing the heterogeneity of EKCs from different Chinese provinces in terms of their positions using a fixed-effect model in the EKC panel estimation, in order to extract the province-specific pollution effects, and to elucidate the factors influencing the province-specific pollution levels. This study focused particularly on the provincial capacity for controlling the pollution levels.

## 3. Materials and Methods

### 3.1. Methodology and Data

This section first overviews the heterogeneity of Chinese provinces in terms of their water pollution levels and the factors influencing them (Table 2, as for the location see

Appendix A). According to the China Statistical Yearbook, in 2020, the industrial discharge of COD per million persons varied from 65 tons in Beijing to 700 tons in Jiangsu; the household discharge of COD was from 1848 tons in Beijing to 11,488 tons in Guangxi; the industrial discharge of ammonia nitrogen was from 2 tons in Beijing to 36 tons in Jiangxi; and the household discharge of ammonia nitrogen was from 94 tons in Tianjin to 1149 tons in Guangxi. The economic factors influencing pollution should be considered; the gross regional product (GRP) per capita in 2010 differed from 127,816 yuan in Beijing to 28,171 yuan in Gansu; the secondary industry's value added as a percentage of GRP (affecting industrial discharges) differed from 46.2% in Fujian to 16.0% in Beijing; and the urban population as a percentage of the total population (affecting household discharges) differed from 89.3% in Shanghai to 35.8% in Tibet. In addition, there are policy priority areas that are designated as key regions for water pollution control by the Chinese government and that are imposed with several regulations to improve water quality. This includes three river (Huai, Hai, and Liao) and three lake (Tai, Chao, and Dianchi) basins (hereafter, 3Rs3Ls) that are spread across 11 provinces (last column of Table 1 [3]). The authors identified the 11 provinces based on Wang et al. [3].

**Table 2.** Water pollution and factors influencing it in Chinese provinces (2020).

| | *codi* | *codh* | *anti* | *anth* | *ypc* | *ind* | *urb* | *3Rs3Ls* |
|---|---|---|---|---|---|---|---|---|
| Beijing | 65 | 1848 | 2 | 119 | 127,816 | 16.0 | 87.5 | * |
| Tianjin | 203 | 2423 | 7 | 94 | 79,377 | 35.1 | 84.7 | * |
| Hebei | 351 | 4823 | 11 | 228 | 37,909 | 38.2 | 60.1 | * |
| Shanxi | 138 | 5386 | 5 | 326 | 40,851 | 43.2 | 62.5 | * |
| Inner Mongolia | 365 | 4290 | 19 | 224 | 56,765 | 40.0 | 67.5 | * |
| Liaoning | 310 | 3893 | 13 | 209 | 46,399 | 37.4 | 72.1 | * |
| Jilin | 392 | 5894 | 15 | 215 | 39,585 | 35.2 | 62.6 | |
| Heilongjiang | 671 | 5672 | 32 | 313 | 33,595 | 25.3 | 65.6 | |
| Shanghai | 346 | 2255 | 8 | 100 | 121,299 | 26.3 | 89.3 | |
| Jiangsu | 700 | 5336 | 30 | 407 | 93,882 | 43.4 | 73.4 | * |
| Zhejiang | 686 | 6299 | 14 | 491 | 78,860 | 40.8 | 72.2 | * |
| Anhui | 268 | 8085 | 16 | 477 | 49,455 | 40.0 | 58.3 | * |
| Fujian | 471 | 10,412 | 18 | 811 | 83,630 | 46.2 | 68.7 | |
| Jiangxi | 459 | 8124 | 36 | 672 | 44,234 | 43.1 | 60.4 | |
| Shandong | 457 | 5140 | 19 | 368 | 56,397 | 39.1 | 63.1 | * |
| Henan | 161 | 5821 | 8 | 347 | 42,522 | 41.0 | 55.4 | * |
| Hubei | 389 | 7674 | 20 | 611 | 56,788 | 37.1 | 62.9 | |
| Hunan | 219 | 7542 | 10 | 728 | 49,459 | 38.4 | 58.8 | |
| Guangdong | 324 | 7162 | 12 | 632 | 68,259 | 39.5 | 74.2 | |
| Guangxi | 312 | 11,488 | 11 | 1149 | 33,915 | 31.9 | 54.2 | |
| Hainan | 437 | 8250 | 11 | 631 | 41,565 | 19.3 | 60.3 | |
| Chongqing | 290 | 4126 | 11 | 511 | 62,176 | 39.8 | 69.5 | |
| Sichuan | 307 | 9413 | 15 | 848 | 45,290 | 36.1 | 56.7 | |
| Guizhou | 122 | 6329 | 17 | 558 | 36,729 | 35.1 | 53.2 | |
| Yunnan | 224 | 5746 | 9 | 435 | 40,611 | 34.2 | 50.0 | |
| Tibet | 49 | 9214 | 3 | 786 | 39,917 | 37.6 | 35.8 | |
| Shaanxi | 239 | 6951 | 8 | 563 | 51,742 | 43.1 | 62.7 | |
| Gansu | 179 | 3998 | 8 | 134 | 28,171 | 31.5 | 52.2 | |
| Qinghai | 274 | 10,516 | 19 | 821 | 38,082 | 38.0 | 60.1 | |
| Ningxia | 436 | 5576 | 17 | 325 | 43,180 | 40.7 | 65.0 | |
| Xinjiang | 424 | 8215 | 23 | 721 | 41,769 | 34.7 | 56.5 | |

Notes: *codi*: Industrial chemical oxygen demand (COD), tons per million persons; *codh*: Household COD, tons per million persons; *anti*: Industrial ammonia nitrogen, tons per million persons; *anth*: Household ammonia nitrogen, tons per million persons; *ypc*: Gross regional product (GRP) per capita, 2010 prices, yuan; *ind*: Secondary industry, percent of GRP; *urb*: Urban population, percent of total population; *3Rs3Ls*: Three river (i.e., Huai, Hai, and Liao) and three lake (i.e., Tai, Chao, and Dianchi) basins; Source: China Statistical Yearbook. * means that the province belongs to *3Rs3Ls*.

This study follows the original form of the EKC: the standard nonlinear model in which water pollution per capita is regressed by income per capita and its square. The original EKC postulates an inverted-U-shaped nexus between pollution per capita and income per capita. Figure 1 depicts the relationships between the water pollution indicators (vertical axis) and the GRP per capita (horizontal axis) in the total 31 sampled Chinese provinces and periods for 2003−2019; they roughly appear to be inverted-U-shaped. The patterns should be further examined by the subsequent econometric tests by controlling the other factors affecting pollution.

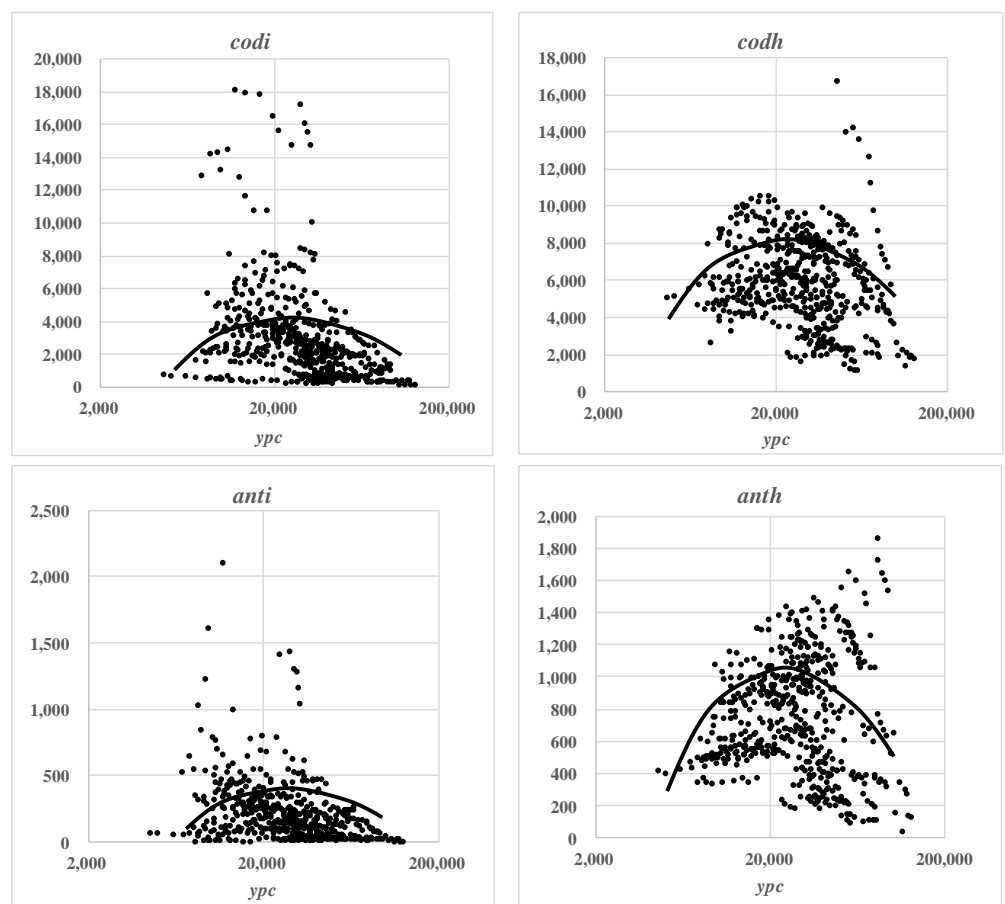

**Figure 1.** Water pollution and GRP per capita. Note: *codi*: Industrial chemical oxygen demand (COD), tons per million persons; *codh*: Household COD, tons per million persons; *anti*: Industrial ammonia nitrogen, tons per million persons; *anth*: Household ammonia nitrogen, tons per million persons; *ypc*: Gross regional product (GRP) per capita, 2010 prices, yuan; Sources: Authors' estimation.

We now turn to econometric approaches. The first specification in Equation (1) applies a fixed-effect model for provincial panel-data estimation in order to explicitly demonstrate the province-specific pollution effects and also run the alternative models in Equations (2) and (3) by replacing the fixed effects with the possible pollution contributors (pollution-control capacity, industrialization, and urbanization) to the province-specific pollution effects. The equations for the estimation are as follows:

$$\ln\left(codi_{it}, codh_{it}, anti_{it}, anth_{it}\right) = \alpha_0 + \alpha_1 \ln ypc_{it} + \alpha_2 \left(\ln ypc_{it}\right)^2 + f_i + f_t + \varepsilon_t \tag{1}$$

$$\ln\left(codi_{it}, anti_{it}\right) = \beta_0 + \beta_1 \ln ypc_{it} + \beta_2 \left(\ln ypc_{it}\right)^2 + \beta_3 \, edu_{it} + \beta_4 \, ind_{it} + f_t + \varepsilon_t \tag{2}$$

$$\ln\left(codh_{it}, anth_{it}\right) = \gamma_0 + \gamma_1 \ln ypc_{it} + \gamma_2 \left(\ln ypc_{it}\right)^2 + \gamma_3 \, edu_{it} + \gamma_4 \, urb_{it} + f_t + \varepsilon_t \tag{3}$$

where the subscript *it* denotes the 31 sampled Chinese provinces for the years 2003−2019, respectively; *codi*, *codh*, *anti*, and *anth* represent the water pollutants: industrial COD, household COD, industrial ammonia nitrogen, and household ammonia nitrogen, respectively, expressed as tons per million persons; *ypc* shows the gross regional product (GRP) per capita in yuan at constant prices in 2010; *edu* denotes the number of higher education graduates per million persons; *ind* shows the secondary industry value added as a percentage of GRP; *urb* represents the urban population as a percentage of the total population; $f_i$ and $f_t$ show a time-invariant country-specific fixed effect and a country-invariant time-specific fixed effect, respectively; $\varepsilon$ denotes a residual error term; $\alpha_{0\ldots2}$, $\beta_{0\ldots4}$, and $\gamma_{0\ldots4}$ represent estimated coefficients; and ln shows a logarithm form, which is set to avoid scaling issues for the water pollutants and GRP per capita. The data source of all the variables is the China Statistical Yearbook. The study constructs a set of panel data for the 31 sample provinces for the period 2003−2019. (This study excluded the year 2020 when the COVID-19 pandemic seriously affected economic activities). The list and descriptive statistics for the variable data are displayed in Tables 3 and 4, respectively.

**Table 3.** List of variables.

| Variables | Description |
| --- | --- |
| Dependent Variable | |
| *codi* | Industrial Chemical Oxygen Demand (COD), ton per million persons, log term |
| *codh* | Household Chemical Oxygen Demand (COD), ton per million persons, log term |
| *anti* | Industrial Ammonia Nitrogen, ton per million persons, log term |
| *anth* | household Ammonia Nitrogen, ton per million persons, log term |
| Explanatory Variables | |
| *ypc* | Gross Domestic Product (GDP) per capita, 2010 prices, RMB, log-term, one-year lagged |
| *edu* | Number of graduate of higher education (regular undergraduate and specialized) per million persons, log-term, ten-year lagged |
| *ind* | Secondary industry, percent of GDP, one-year lagged |
| *urb* | Urban population, percent of total population, one-year lagged |

Sources: Authors' description.

**Table 4.** Descriptive statistics.

| Variables | Obs. | Median | Std. Dev. | Min. | Max |
| --- | --- | --- | --- | --- | --- |
| Dependent Variable | | | | | |
| *codi* | 527 | 7.700 | 1.046 | 4.159 | 9.800 |
| *codh* | 527 | 8.672 | 0.458 | 6.984 | 9.724 |
| *anti* | 527 | 7.648 | 1.295 | 0.000 | 7.648 |
| *anth* | 527 | 6.583 | 0.584 | 3.689 | 7.532 |
| Explanatory Variables | | | | | |
| *ypc* | 527 | 10.298 | 0.620 | 8.435 | 11.761 |
| *edu* | 527 | 8.351 | 0.536 | 6.463 | 9.214 |
| *ind* | 527 | 42.340 | 8.895 | 15.989 | 63.254 |
| *urb* | 519 | 50.970 | 14.626 | 22.198 | 89.600 |

Sources: Authors' calculation.

The notes on the specifications of the estimation models in (1), (2), and (3) are required for an additional description as follows. Equation (1) applies a fixed-effect model, represented by $f_i$ and $f_t$, for provincial panel-data estimation. The Hausman test is generally used for choosing between a fixed-effect model and a random effect model [25]. This study, however, focused on demonstrating province-specific pollution effects explicitly; time-specific factors such as economic fluctuations due to external shocks, such as the Asian financial crises in 1997–1998 and the global financial crises in 2008–2009, were considered. In addition, adopting the fixed-effect model contributes to alleviating the endogeneity problem by absorbing the unobserved time-invariant heterogeneity among the sample provinces.

The estimation sets Beijing as the benchmark province for extracting the province-specific pollution effects, because Beijing shows the best performance in water pollution control (Table 1). The significantly positive coefficient of the province-specific fixed effect suggests that the water pollution in the particular province is more serious than that in Beijing. The ordinary hypothesis of the EKC postulating the inverted-U-shaped path between water pollution and GRP per capita would be verified if $\alpha_1$, $\beta_1$, $\gamma_1 > 0$ and $\alpha_2$, $\beta_2$, $\gamma_2 < 0$ are significant with reasonable levels of turning points.

Equations (2) and (3) represent the alternative models for industrial discharges and household discharges, respectively. Equation (2) replaces the province-specific fixed effects with the possible pollution contributors of the fixed effects: pollution-control capacity (*edu*) and industrialization (*ind*). Equation (3) replaces them with pollution-control capacity (*edu*) and urbanization (*urb*). This study uses the number of graduates of higher education (*edu*) to represent the capacity to control pollution because the pollution controllability depends highly on human resources and capital in order to address the pollution level in each province. We attempted to apply the variable of treatment plant capacity as the capacity to control pollution. Appendix B showed that the daily treatment capacity of sewage (*tcs*) was positively correlated with water pollution. It suggests that the treatment plant capacities have only chased after the pollution and thus are equipped with no significant power for pollution control. The importance of human capital in controlling environmental pollution has been studied widely [26–28]. The adoption of industrialization (*ind*) and urbanization (*urb*) is based on Liu et al.'s study [22]; secondary industry output can be a main indicator for industrial water use, and urban population can be an indicator for household water use. We considered the adoption of the indicators directly measuring industrial and household waste water discharge rates. However, these data are available only for 2003–2015 (the Government has stopped publishing these data since 2016). Another problem is that the level of waste water discharge itself can be affected by pollution control variables, which leads to a multicollinearity problem. Thus, this study alternatively used the indirect indicators, including industrialization and urbanization. Appendix C attempted the estimation using the direct indicators and still found negative coefficients for the pollution control variable (*edu*), despite some instabilities in the coefficients due to the problems above. No multicollinearity problem exists in the regressors' combinations in Equations (2) and (3), namely, (*ypc*, *edu*, *ind*) and (*ypc*, *edu*, *urb*). This is because the variance inflation factors (VIFs), reflecting the level of collinearity between the regressors, indicate lower values than the criteria of collinearity (10 points) in each equation. The VIF values of *ypc*, *edu*, and *ind* in Equation (2) are 2.793, 2.765, and 1.017, respectively, and those of *ypc*, *edu*, and *urb* in Equation (3) are 5.417, 2.737, and 4.111, respectively, according to the authors' estimation. The pollution-control capacity (*edu*) is expected to impart a negative coefficient for water pollution because the higher capacity enables the mitigation of pollution. The coefficients of industrialization (*ind*) and urbanization (*urb*), which deteriorate water quality, are supposed to be positive in the respective equations.

The explanatory variables in Equations (1)–(3), *ypc*, *ind*, and *urb* were lagged by one year. This helps avoid reverse causality in the model specifications, including the endogenous interaction between the dependent and independent variables. For the pollution-control capacity (*edu*), a 10-year lag was applied because it takes a long time for graduates of higher education to be trained for capacity building for pollution control. Figure 2 displays the magnitudes of negative coefficients for the pollution-control capacity (*edu*) using time series lag patterns from Equations (2) and (3), estimated for each water pollutant; the impacts of the capacity on pollution levels are negatively maximized around the 10-year lag, though the impact sizes differ according to the difference in the effects by their treatment processes.

This study applies the ordinary least squares (OLS) estimator and the Poisson pseudo-maximum likelihood (PPML) estimator for the estimations. The PPML estimator was selected because the sample data with heterogeneity in the provincial properties would be plagued by heteroskedasticity and autocorrelation; in such cases, the OLS estimator leads

to bias and inconsistency in the estimates. The PPML estimator corrects for heteroscedastic error structure across panels and autocorrelation with panels, as Silva and Tenreyro [29] and Kareem et al. [30] suggest. Therefore, these two estimators are applied to ensure the robustness of the estimations. We used EViews (version 12) (IHS Global Inc., CA, USA) for processing the data and estimations.

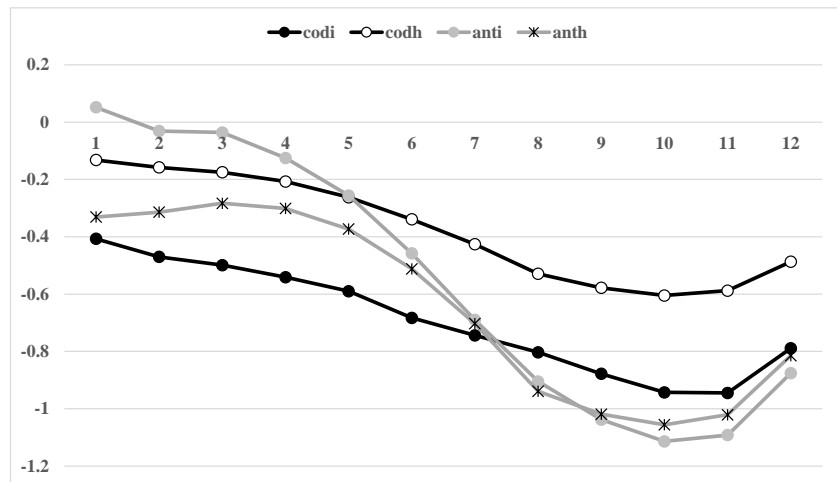

**Figure 2.** Lag pattern of pollution-control capacity. Note: The vertical line denotes the coefficient value of pollution-control capacity (*edu*), defined as the number of graduates of higher education per million persons. The horizontal line indicates yearly time lags of the *edu* variable: $edu_{-1}$, $edu_{-2}$, $edu_{-3}$ ... $edu_{-n}$. Sources: Authors' estimation.

### 3.2. Panel Unit Root and Cointegration Tests

For the subsequent estimation, we investigated the stationary property of the panel data by utilizing panel unit root tests, and if necessary, a panel cointegration test for a set of variables' data. The panel unit root tests were first conducted on the null hypothesis such that a level and/or the first difference of the individual data have a unit root. In cases where the unit root tests reveal that each variable's data are not stationary in the level, but stationary in the first difference, a set of variables' data correspond to the case of $I(1)$; this can be further examined using a co-integration test for the "level" data. If a set of variables' data are identified to have a co-integration, the use of the "level" data is justified for model estimation.

For the panel unit root tests, this study applied the Levin, Lin, and Chu test [31] as a common unit root test, and the Fisher-ADF and Fisher-PP tests [32,33] as individual unit root tests. The common unit root test assumes a common unit root process across cross-sections, and the individual unit root test allows for individual unit root processes that vary across cross sections. For a panel co-integration test, the study used the Pedroni residual co-integration test (developed by Pedroni [34]). All of the test equations contained an individual intercept and trend, with the lag length being an automatic selection.

Table 5 presents the test results: the common unit root test rejects the null hypothesis of a unit root at the conventional significance levels for all of the variables. However, the individual tests do not reject a unit root in their levels, except *edu*, while rejecting it in their first differences; therefore, the variables almost follow the case of $I(1)$. The panel co-integration test was conducted further on the combinations of variables in Equations (2) and (3). The panel PP and ADF tests suggested that the level series of a set of variables' data are co-integrated in the respective combinations. Thus, this study utilizes the level data for the estimation.

**Table 5.** Panel unit root tests.

| | Unit Root Test | | Panel Cointegration Test | |
|---|---|---|---|---|
| | Level | 1st Difference | | |
| | Levin, Lin and Chu | | | |
| *codi* | −3.959 *** | - | | |
| *codh* | −3.445 *** | - | | |
| *anti* | −3.277 *** | - | | |
| *anth* | −3.089 *** | - | - | |
| *ypc* | −2.223 ** | - | | |
| $ypc^2$ | −1.365 * | - | | |
| *edu* | −2.895 *** | - | | |
| *ind* | −3.290 *** | - | | |
| | Fisher-ADF | | Panal ADF | Panel PP |
| *codi* | 30.454 | 178.523 *** | Group of *codi* | |
| *codh* | 33.619 | 219.161 *** | | |
| *anti* | 37.407 | 200.169 *** | −3.661 *** | −0.863 *** |
| *anth* | 9.257 | 181.955 *** | | |
| *ypc* | 21.936 | 145.492 *** | Group of *codh* | |
| $ypc^2$ | 17.626 | 149.354 *** | | |
| *edu* | 79.095 * | 113.663 *** | −4.102 *** | −4.037 *** |
| *ind* | 70.484 | 201.233 *** | | |
| | Fisher-PP | | Panal ADF | Panel PP |
| *codi* | 26.552 | 265.096 *** | Group of *anti* | |
| *codh* | 36.131 | 290.202 *** | | |
| *anti* | 33.946 | 277.300 *** | −3.248 *** | −2.121 ** |
| *anth* | 4.284 | 204.084 *** | | |
| *ypc* | 29.065 | 150.792 *** | Group of *anth* | |
| $ypc^2$ | 20.094 | 141.915 *** | | |
| *edu* | 425.975 *** | 131.504 *** | −3.223 *** | −2.116 ** |
| *ind* | 60.512 | 264.166 *** | | |

Note: ***, **, and * denote statistical significance at 99, 95, and 90% level, respectively. Sources: Authors' estimation.

## 4. Results and Discussion

Tables 6–9 present the results of the OLS and PPML estimations in the form of a log-link function for industrial and household COD, and industrial and household ammonia nitrogen, respectively. Columns (i) and (ii) display the outcomes of the fixed-effect models, and columns (iii) and (iv) present the results of the alternative models containing pollution-control capacity (*edu*) and industrialization (*ind*) for industrial discharges, and pollution-control capacity (*edu*) and urbanization (*urb*) for household discharges, instead of the fixed effects. Both the OLS and PPML estimations show similar results in the sign and significance of each coefficient; therefore, the subsequent description focuses on the result of the PPML estimations that adjust heteroskedasticity and autocorrelation. The findings from the estimation results are summarized as follows.

**Table 6.** Estimation results: industrial COD (*codi*).

| Estimation Methodology | (i) OLS | (ii) PPML | (iii) OLS | (iv) PPML |
|---|---|---|---|---|
| *ypc* | 2.603 ** | 2.796 ** | 12.333 ** | 13.404 ** |
| | (2.336) | (2.229) | (2.692) | (2.216) |
| *ypc²* | −0.104 * | −0.112 * | −0.567 ** | −0.618 ** |
| | (−1.747) | (−1.663) | (−2.653) | (−2.207) |
| *ind* | | | 0.019 *** | 0.020 *** |
| | | | (5.994) | (3.262) |
| *edu* | | | −0.940 *** | −0.943 *** |
| | | | (−17.758) | (−6.774) |
| Dummy for fixed effect | | | | |
| Tianjin | 1.856 *** | 1.825 *** | | |
| Hebei | 2.484 *** | 2.468 *** | | |
| Shanxi | 2.549 *** | 2.543 *** | | |
| Inner Mongolia | 2.947 *** | 2.970 *** | | |
| Liaoning | 2.654 *** | 2.676 *** | | |
| Jilin | 2.903 *** | 2.924 *** | | |
| Heilongjiang | 2.646 *** | 2.680 *** | | |
| Shanghai | 1.363 *** | 1.381 *** | | |
| Jiangsu | 2.562 *** | 2.603 *** | | |
| Zhejiang | 2.618 *** | 2.630 *** | | |
| Anhui | 2.239 *** | 2.276 *** | | |
| Fujian | 2.262 *** | 2.298 *** | | |
| Jiangxi | 2.705 *** | 2.775 *** | | |
| Shandong | 2.207 *** | 2.227 *** | | |
| Henan | 2.273 *** | 2.262 *** | | |
| Hubei | 2.313 *** | 2.316 *** | | |
| Hunan | 2.691 *** | 2.721 *** | | |
| Guangdong | 2.155 *** | 2.178 *** | | |
| Guangxi | 3.427 *** | 3.419 *** | | |
| Hainan | 2.001 *** | 2.067 *** | | |
| Chongqing | 2.443 *** | 2.458 *** | | |
| Sichuan | 2.466 *** | 2.493 *** | | |
| Guizhou | 1.662 *** | 1.701 *** | | |
| Yunnan | 2.603 *** | 2.632 *** | | |
| Tibet | 0.918 ** | 1.044 ** | | |
| Shaanxi | 2.547 *** | 2.548 *** | | |
| Gansu | 2.773 *** | 2.840 *** | | |
| Qinghai | 3.089 *** | 3.089 *** | | |
| Ningxia | 4.140 *** | 4.157 *** | | |
| Xinjiang | 3.429 *** | 3.441 *** | | |
| Turning Point (*ypc*) | 12.456 | 12.457 | 10.872 | 10.842 |
| Cross-sections | 31 | 31 | 31 | 31 |
| Periods | 2004–2019 | 2004–2019 | 2013–2019 | 2013–2019 |
| Total observations | 496 | 496 | 217 | 217 |

Note: ***, **, and * denote statistical significance at 99, 95, and 90% level, respectively. T-statistics are in the parentheses. Sources: Authors' estimation.

**Table 7.** Estimation results: household COD (*codh*).

| Estimation Methodology | (i) OLS | (ii) PPML | (iii) OLS | (iv) PPML |
|---|---|---|---|---|
| *ypc* | 3.461 *** | 3.518 *** | 5.497 *** | 5.615 * |
|  | (5.815) | (5.254) | (8.746) | (1.949) |
| $ypc^2$ | −0.151 *** | −0.155 *** | −0.259 *** | −0.262 ** |
|  | (−4.731) | (−4.222) | (−8.681) | (−1.969) |
| *urb* |  |  | 0.012 * | 0.009 ** |
|  |  |  | (2.357) | (2.123) |
| *edu* |  |  | −0.615 *** | −0.605 *** |
|  |  |  | (−21.443) | (−8.886) |
| dummy for fixed effect |  |  |  |  |
| Tianjin | 0.396 *** | 0.380 *** |  |  |
| Hebei | 0.203 | 0.186 |  |  |
| Shanxi | 0.524 *** | 0.500 ** |  |  |
| Inner Mongolia | 0.391 *** | 0.363 ** |  |  |
| Liaoning | 0.597 *** | 0.573 *** |  |  |
| Jilin | 0.734 *** | 0.702 *** |  |  |
| Heilongjiang | 0.969 *** | 0.945 *** |  |  |
| Shanghai | 0.512 *** | 0.503 *** |  |  |
| Jiangsu | 0.571 *** | 0.570 *** |  |  |
| Zhejiang | 0.295 *** | 0.289 *** |  |  |
| Anhui | 0.721 *** | 0.702 *** |  |  |
| Fujian | 0.789 *** | 0.783 *** |  |  |
| Jiangxi | 1.018 *** | 0.995 *** |  |  |
| Shandong | 0.107 | 0.095 |  |  |
| Henan | 0.218 | 0.198 |  |  |
| Hubei | 0.812 *** | 0.794 *** |  |  |
| Hunan | 0.940 *** | 0.915 *** |  |  |
| Guangdong | 0.648 *** | 0.645 *** |  |  |
| Guangxi | 1.115 *** | 1.090 *** |  |  |
| Hainan | 1.017 *** | 0.992 *** |  |  |
| Chongqing | 0.276 * | 0.241 |  |  |
| Sichuan | 0.731 *** | 0.705 *** |  |  |
| Guizhou | 0.817 *** | 0.785 *** |  |  |
| Yunnan | 0.438 ** | 0.405 * |  |  |
| Tibet | 0.795 *** | 0.774 *** |  |  |
| Shaanxi | 0.414 ** | 0.389 ** |  |  |
| Gansu | 0.530 ** | 0.493 ** |  |  |
| Qinghai | 0.746 *** | 0.720 *** |  |  |
| Ningxia | 0.312 * | 0.298 |  |  |
| Xinjiang | 0.639 *** | 0.628 *** |  |  |
| Turning Point (*ypc*) | 11.452 | 11.356 | 10.612 | 10.713 |
| Cross-sections | 31 | 31 | 31 | 31 |
| Periods | 2004–2019 | 2004–2019 | 2013–2019 | 2013–2019 |
| Total observations | 496 | 496 | 217 | 217 |

Note: ***, **, and * denote statistical significance at 99, 95, and 90% level, respectively. T-statistics are in the parentheses. Sources: Authors' estimation.

**Table 8.** Estimation results: industrial ammonia nitrogen (*anti*).

| Estimation Methodology | (i) OLS | (ii) PPML | (iii) OLS | (iv) PPML |
|---|---|---|---|---|
| *ypc* | 4.038 *** (2.647) | 6.935 *** (3.318) | 17.671 ** (2.109) | 21.456 ** (2.383) |
| *ypc²* | −0.178 ** (−2.177) | −0.317 *** (−2.804) | −0.811 ** (−2.091) | −0.989 ** (−2.375) |
| *ind* | | | 0.023 ** (2.549) | 0.024 *** (2.972) |
| *edu* | | | −1.098 *** (−6.128) | −1.114 *** (−6.737) |
| Dummy for fixed effect | | | | |
| Tianjin | 2.283 *** | 2.172 *** | | |
| Hebei | 2.556 *** | 2.484 *** | | |
| Shanxi | 2.645 *** | 2.570 *** | | |
| Inner Mongolia | 2.891 *** | 2.906 *** | | |
| Liaoning | 2.473 *** | 2.446 *** | | |
| Jilin | 2.366 *** | 2.425 *** | | |
| Heilongjiang | 2.455 *** | 2.459 *** | | |
| Shanghai | 1.631 *** | 1.809 *** | | |
| Jiangsu | 2.546 *** | 2.688 *** | | |
| Zhejiang | 2.515 *** | 2.469 *** | | |
| Anhui | 2.432 *** | 2.418 *** | | |
| Fujian | 2.241 *** | 2.270 *** | | |
| Jiangxi | 2.707 *** | 2.803 *** | | |
| Shandong | 2.148 *** | 2.155 *** | | |
| Henan | 2.357 *** | 2.260 *** | | |
| Hubei | 2.638 *** | 2.584 *** | | |
| Hunan | 3.151 *** | 3.142 *** | | |
| Guangdong | 1.726 *** | 1.778 *** | | |
| Guangxi | 2.945 *** | 2.929 *** | | |
| Hainan | 1.709 *** | 1.811 *** | | |
| Chongqing | 2.374 *** | 2.351 *** | | |
| Sichuan | 2.169 *** | 2.198 *** | | |
| Guizhou | 1.664 *** | 1.791 *** | | |
| Yunnan | 1.862 *** | 1.926 *** | | |
| Tibet | −0.336 | 0.108 | | |
| Shaanxi | 2.168 *** | 2.151 *** | | |
| Gansu | 3.257 *** | 3.223 *** | | |
| Qinghai | 2.716 *** | 2.757 *** | | |
| Ningxia | 4.052 *** | 4.037 *** | | |
| Xinjiang | 2.999 *** | 3.039 *** | | |
| Turning Point (*ypc*) | 11.327 | 10.922 | 10.891 | 10.844 |
| Cross-sections | 31 | 31 | 31 | 31 |
| Periods | 2004–2019 | 2004–2019 | 2013–2019 | 2013–2019 |
| Total observations | 496 | 496 | 217 | 217 |

Note: *** and ** denote statistical significance at 99 and 95% level, respectively. T-statistics are in the parentheses. Sources: Authors' estimation.

**Table 9.** Estimation results: household ammonia nitrogen (*anth*).

| Estimation Methodology | (i) OLS | (ii) PPML | (iii) OLS | (iv) PPML |
|---|---|---|---|---|
| *ypc* | 5.780 *** (8.026) | 6.078 *** (7.475) | 8.284 ** (3.111) | 8.609 ** (1.998) |
| *ypc²* | −0.256 *** (−6.621) | −0.270 *** (−6.343) | −0.398 ** (−3.088) | −0.413 ** (−2.058) |
| *ind* | | | 0.026 ** (2.954) | 0.026 *** (3.713) |
| *edu* | | | −1.054 *** (−19.299) | −1.056 *** (−12.278) |
| Dummy for fixed effect | | | | |
| Tianjin | 0.381 *** | 0.361 ** | | |
| Hebei | 0.486 ** | 0.521 ** | | |
| Shanxi | 0.874 *** | 0.900 *** | | |
| Inner Mongolia | 0.716 *** | 0.702 *** | | |
| Liaoning | 0.951 *** | 0.957 *** | | |
| Jilin | 0.964 *** | 0.973 *** | | |
| Heilongjiang | 1.191 *** | 1.219 *** | | |
| Shanghai | 0.896 *** | 0.922 *** | | |
| Jiangsu | 0.562 *** | 0.596 *** | | |
| Zhejiang | 0.334 *** | 0.354 ** | | |
| Anhui | 0.765 *** | 0.794 *** | | |
| Fujian | 0.764 *** | 0.790 *** | | |
| Jiangxi | 1.014 *** | 1.052 *** | | |
| Shandong | 0.378 ** | 0.400 ** | | |
| Henan | 0.530 ** | 0.557 ** | | |
| Hubei | 0.939 *** | 0.968 *** | | |
| Hunan | 1.047 *** | 1.078 *** | | |
| Guangdong | 0.784 *** | 0.811 *** | | |
| Guangxi | 1.100 *** | 1.140 *** | | |
| Hainan | 1.127 *** | 1.159 *** | | |
| Chongqing | 0.534 *** | 0.533 ** | | |
| Sichuan | 0.828 *** | 0.864 *** | | |
| Guizhou | 1.022 *** | 1.073 *** | | |
| Yunnan | 0.613 ** | 0.641 ** | | |
| Tibet | 0.992 *** | 1.043 *** | | |
| Shaanxi | 0.654 *** | 0.675 *** | | |
| Gansu | 0.836 *** | 0.862 *** | | |
| Qinghai | 1.219 *** | 1.253 *** | | |
| Ningxia | 0.881 *** | 0.920 *** | | |
| Xinjiang | 1.098 *** | 1.152 *** | | |
| Turning Point (*ypc*) | 11.294 | 11.256 | 10.412 | 10.435 |
| Cross-sections | 31 | 31 | 31 | 31 |
| Periods | 2004–2019 | 2004–2019 | 2013–2019 | 2013–2019 |
| Total observations | 496 | 496 | 217 | 217 |

Note: *** and ** denote statistical significance at 99 and 95% level, respectively. T-statistics are in the parentheses. Sources: Authors' estimation.

### 4.1. EKC Identification by Fixed-Effect Model

First, the EKC hypothesis, which assumes the inverted-U-shaped relationship between water pollution level and GRP per capita, was confirmed in all the water pollutants from Tables 6–9 and in all the estimations from columns (i)–(iv). They were confirmed using the estimation results; the coefficients of the GRP per capita were significantly positive, and those of its square were significantly negative. The turning points fell within the reasonable ranges of GRP per capita between its minimum and maximum levels in the samples shown in Table 4, except for the estimations in columns (i) and (ii) in Table 6 (the turning point was computed using $-\alpha_1/2\alpha_2$, $-\beta_1/2\beta_2$, or $-\gamma_1/2\gamma_2$ in the Equations (1)–(3)). The finding of the inverted-U-shaped EKC in Chinese provinces in this study is consistent with previous

studies: Zhang et al. [2], Zhao et al. [18], Li et al. [19], Jayanthakumaran and Liu [20], and Shen [21] in Table 2. The main research focus in this study was, however, the provincial EKC positions rather than their shapes, as in the subsequent description.

### 4.2. Extraction of Provincial-Specific Pollution Effect

Second, the fixed-effect models in columns (i) and (ii) identified the positive coefficients as the province-specific fixed effects at conventional significant levels, in all the provinces for industrial COD in Table 6 and for household ammonia nitrogen in Table 9, and in the majority of provinces for household COD (except Hebei, Shandong, Henan, Chongqing, and Ningxia) in Table 7 and for industrial ammonia nitrogen (except Tibet) in Table 8. The positive provincial fixed effects mean that the provincial EKCs are located above Beijing, which is the benchmark, suggesting that the province-specific pollution effects (not affected by the provincial income level on the EKC) are larger than those in Beijing. These results are in line with the simple observations on water pollution per capita in all the provinces in Table 1. The degree of water pollution was indicated by the magnitude of the coefficients of provincial fixed effects: the industrial COD in Tianjin via the PPML estimation (column (ii) in Table 6), for instance, was exp. (1.825) = 6.203 times larger than that in Beijing. The provincial fixed effects also revealed that the pollution levels in the policy propriety areas (3Rs3Ls), shown in Table 1, were not necessarily higher than the average levels among the 31 provinces for all water pollutants, thereby implying that the government policies have controlled the water pollution in the priority areas.

### 4.3. Re-Estimation Results of Alternative EKC Model

Third, in the alternative model containing pollution-control capacity (*edu*) and industrialization (*ind*)/urbanization (*urb*) in columns (iii) and (iv), respectively, the coefficients of *edu* were significantly negative in all the pollutants and estimations in Tables 6–9; those of *ind* for industrial discharges in Tables 6 and 8 and those of *urb* for household discharges in Tables 7 and 9 were significantly positive in all estimations. These results are in line with the hypothesis of Liu et al. [22], stating that the secondary industry output and urban population can be the main indicators of industrial and household water use, respectively. More importantly, the negative coefficients of *edu* for all pollutants suggest that the pollution-control capacity had, indeed, affected the provincial pollution levels and that the heterogeneity of provincial pollution could be explained by the differences in the provincial pollution-control capacity. The joint estimation outcomes of the province-specific pollution effects and the workability of pollution-control capacity lead to a question regarding the quantitative contributions of provincial capacity shortage to pollution control at the provincial pollution level.

### 4.4. Factor Analysis on Pollution-Control Capacity

We quantified the contributions of the provincial pollution-control capacity to the province-specific pollution effects (also based on the PPML estimation). Tables 10 and 11 present the analytical outcomes for COD and ammonia nitrogen discharges, respectively. Columns (a) and (b) repeat the provincial fixed effects (only significant coefficients) in Tables 6–9, representing the province-specific pollution from industrial and household discharges, respectively; column (c) presents the period average of provincial pollution-control capacity indicators (*edu*); column (d) computes the *edu* deviations from that of Beijing (the benchmark); columns (e) and (f) indicate the *edu* contributions to provincial industrial and household discharges, respectively, by multiplying the *edu* deviations with the estimated *edu* coefficients in Tables 6–9; and columns (g) and (h) demonstrate the *edu* contribution ratios to provincial industrial and household pollution by dividing columns (e) and (f) by columns (a) and (b), respectively.

**Table 10.** Provincial pollution and pollution-control capacity (COD).

| cod | Fixed Effect codi (a) | Fixed Effect codh (b) | edu (c) | (c) − Benchmark (d) | (d) × −0.943 codi (e) | (d) × −0.605 codh (f) | (e)/(a) codi (g) | (f)/(b) codh (h) |
|---|---|---|---|---|---|---|---|---|
| Tianjin | 1.825 | 0.380 | 8.982 | 0.080 | −0.075 | −0.048 | −0.041 | −0.127 |
| Hebei | 2.468 | - | 8.225 | −0.677 | 0.639 | - | 0.259 | - |
| Shanxi | 2.543 | - | 8.295 | −0.607 | 0.573 | - | 0.225 | - |
| Inner Mongolia | 2.970 | - | 8.077 | −0.825 | 0.778 | - | 0.262 | - |
| Liaoning | 2.676 | 0.573 | 8.459 | −0.444 | 0.418 | 0.268 | 0.156 | 0.468 |
| Jilin | 2.924 | 0.702 | 8.447 | −0.455 | 0.429 | 0.275 | 0.147 | 0.392 |
| Heilongjiang | 2.680 | 0.945 | 8.373 | −0.529 | 0.499 | 0.320 | 0.186 | 0.339 |
| Shanghai | 1.381 | 0.503 | 8.594 | −0.309 | 0.291 | 0.187 | 0.211 | 0.371 |
| Jiangsu | 2.603 | 0.570 | 8.447 | −0.455 | 0.429 | 0.275 | 0.165 | 0.482 |
| Zhejiang | 2.630 | 0.289 | 8.213 | −0.690 | 0.651 | 0.417 | 0.247 | 1.443 |
| Anhui | 2.276 | 0.702 | 8.147 | −0.755 | 0.712 | 0.457 | 0.313 | 0.650 |
| Fujian | 2.298 | 0.783 | 8.184 | −0.719 | 0.678 | 0.435 | 0.295 | 0.555 |
| Jiangxi | 2.775 | 0.995 | 8.370 | −0.532 | 0.502 | 0.322 | 0.181 | 0.323 |
| Shandong | 2.227 | - | 8.285 | −0.618 | 0.583 | - | 0.262 | - |
| Henan | 2.262 | - | 8.149 | −0.753 | 0.710 | - | 0.314 | - |
| Hubei | 2.316 | 0.794 | 8.559 | −0.343 | 0.324 | 0.207 | 0.140 | 0.261 |
| Hunan | 2.721 | 0.915 | 8.212 | −0.691 | 0.651 | 0.418 | 0.239 | 0.456 |
| Guangdong | 2.178 | 0.645 | 7.980 | −0.922 | 0.870 | 0.558 | 0.399 | 0.865 |
| Guangxi | 3.419 | 1.090 | 7.878 | −1.025 | 0.967 | 0.620 | 0.283 | 0.568 |
| Hainan | 2.067 | 0.992 | 8.053 | −0.849 | 0.801 | 0.513 | 0.388 | 0.518 |
| Chongqing | 2.458 | - | 8.286 | −0.616 | 0.581 | - | 0.236 | - |
| Sichuan | 2.493 | 0.705 | 8.019 | −0.884 | 0.834 | 0.534 | 0.334 | 0.757 |
| Guizhou | 1.701 | 0.785 | 7.638 | −1.265 | 1.193 | 0.765 | 0.701 | 0.975 |
| Yunnan | 2.632 | - | 7.651 | −1.251 | 1.180 | - | 0.448 | - |
| Tibet | - | 0.774 | 7.609 | −1.294 | - | 0.782 | - | 1.011 |
| Shaanxi | 2.548 | - | 8.658 | −0.244 | 0.230 | - | 0.090 | - |
| Gansu | 2.840 | - | 8.097 | −0.805 | 0.759 | - | 0.267 | - |
| Qinghai | 3.089 | 0.720 | 7.569 | −1.334 | 1.258 | 0.806 | 0.407 | 1.120 |
| Ningxia | 4.157 | - | 7.911 | −0.992 | 0.936 | - | 0.225 | - |
| Xinjiang | 3.441 | 0.628 | 7.839 | −1.064 | 1.003 | 0.643 | 0.292 | 1.024 |

Sources: Authors' estimation.

**Table 11.** Provincial pollution and pollution-control capacity (ammonia nitrogen).

| ant | Fixed Effect anti (a) | Fixed Effect anth (b) | edu (c) | (c) − Benchmark (d) | (d) × −1.114 anti (e) | (d) × −1.056 anth (f) | (e)/(a) anti (g) | (f)/(b) anth (h) |
|---|---|---|---|---|---|---|---|---|
| Tianjin | 2.172 | - | 8.982 | 0.080 | −0.089 | - | −0.041 | - |
| Hebei | 2.484 | - | 8.225 | −0.677 | 0.755 | - | 0.304 | - |
| Shanxi | 2.570 | 0.900 | 8.295 | −0.607 | 0.677 | 0.641 | 0.263 | 0.713 |
| Inner Mongolia | 2.906 | 0.702 | 8.077 | −0.825 | 0.919 | 0.872 | 0.316 | 1.242 |
| Liaoning | 2.446 | 0.957 | 8.459 | −0.444 | 0.494 | 0.469 | 0.202 | 0.490 |
| Jilin | 2.425 | 0.973 | 8.447 | −0.455 | 0.507 | 0.481 | 0.209 | 0.494 |
| Heilongjiang | 2.459 | 1.219 | 8.373 | −0.529 | 0.590 | 0.559 | 0.240 | 0.459 |
| Shanghai | 1.809 | 0.922 | 8.594 | −0.309 | 0.344 | 0.326 | 0.190 | 0.354 |
| Jiangsu | 2.688 | 0.596 | 8.447 | −0.455 | 0.507 | 0.481 | 0.189 | 0.806 |
| Zhejiang | 2.469 | - | 8.213 | −0.690 | 0.769 | - | 0.311 | - |
| Anhui | 2.418 | 0.794 | 8.147 | −0.755 | 0.841 | 0.798 | 0.348 | 1.005 |
| Fujian | 2.270 | 0.790 | 8.184 | −0.719 | 0.801 | 0.759 | 0.353 | 0.961 |
| Jiangxi | 2.803 | 1.052 | 8.370 | −0.532 | 0.593 | 0.562 | 0.212 | 0.534 |
| Shandong | 2.155 | - | 8.285 | −0.618 | 0.688 | - | 0.319 | - |
| Henan | 2.260 | - | 8.149 | −0.753 | 0.839 | - | 0.371 | - |
| Hubei | 2.584 | 0.968 | 8.559 | −0.343 | 0.382 | 0.362 | 0.148 | 0.374 |
| Hunan | 3.142 | 1.078 | 8.212 | −0.691 | 0.769 | 0.730 | 0.245 | 0.677 |
| Guangdong | 1.778 | 0.811 | 7.980 | −0.922 | 1.028 | 0.974 | 0.578 | 1.201 |

**Table 11.** *Cont.*

| ant | Fixed Effect anti (a) | anth (b) | edu (c) | (c) − Benchmark (d) | (d) × −1.114 anti (e) | (d) × −1.056 anth (f) | (e)/(a) anti (g) | (f)/(b) anth (h) |
|---|---|---|---|---|---|---|---|---|
| Guangxi | 2.929 | 1.140 | 7.878 | −1.025 | 1.142 | 1.083 | 0.390 | 0.949 |
| Hainan | 1.811 | 1.159 | 8.053 | −0.849 | 0.946 | 0.897 | 0.522 | 0.774 |
| Chongqing | 2.351 | - | 8.286 | −0.616 | 0.686 | - | 0.292 | - |
| Sichuan | 2.198 | 0.864 | 8.019 | −0.884 | 0.985 | 0.934 | 0.448 | 1.080 |
| Guizhou | 1.791 | 1.073 | 7.638 | −1.265 | 1.409 | 1.336 | 0.787 | 1.245 |
| Yunnan | 1.926 | - | 7.651 | −1.251 | 1.394 | - | 0.724 | - |
| Tibet | - | 1.043 | 7.609 | −1.294 | - | 1.367 | - | 1.310 |
| Shaanxi | 2.151 | 0.675 | 8.658 | −0.244 | 0.272 | 0.258 | 0.127 | 0.382 |
| Gansu | 3.223 | 0.862 | 8.097 | −0.805 | 0.897 | 0.851 | 0.278 | 0.987 |
| Qinghai | 2.757 | 1.253 | 7.569 | −1.334 | 1.486 | 1.409 | 0.539 | 1.125 |
| Ningxia | 4.037 | 0.920 | 7.911 | −0.992 | 1.105 | 1.048 | 0.274 | 1.138 |
| Xinjiang | 3.039 | 1.152 | 7.839 | −1.064 | 1.185 | 1.124 | 0.390 | 0.975 |

Sources: Authors' estimation.

Figure 3 demonstrates that the average *edu* contribution ratios among the total provinces except those with insignificant fixed effects were 0.263 for industrial COD, 0.623 for household COD, 0.329 for industrial ammonia nitrogen, and 0.838 for household ammonia nitrogen. Therefore, the capacity shortage for pollution control as one of the pollution factors accounts for (a) about 30% of industrial COD and industrial ammonia nitrogen; (b) about 60% of household COD; and (c) about 80% of household ammonia nitrogen. This highlights the significance of building the capacity for water pollution control by developing human resources and training them. Capacity building contributes to the mitigation of water pollution through various channels by enhancing environmental awareness (e.g., Niu et al. [35]), developing environmental technologies (e.g., Zhao et al. [18], Aboelmaged and Hashem [36]), and improving the regulatory powers and governance of environmental policies (e.g., Cai et al. [1], Liu et al. [22]).

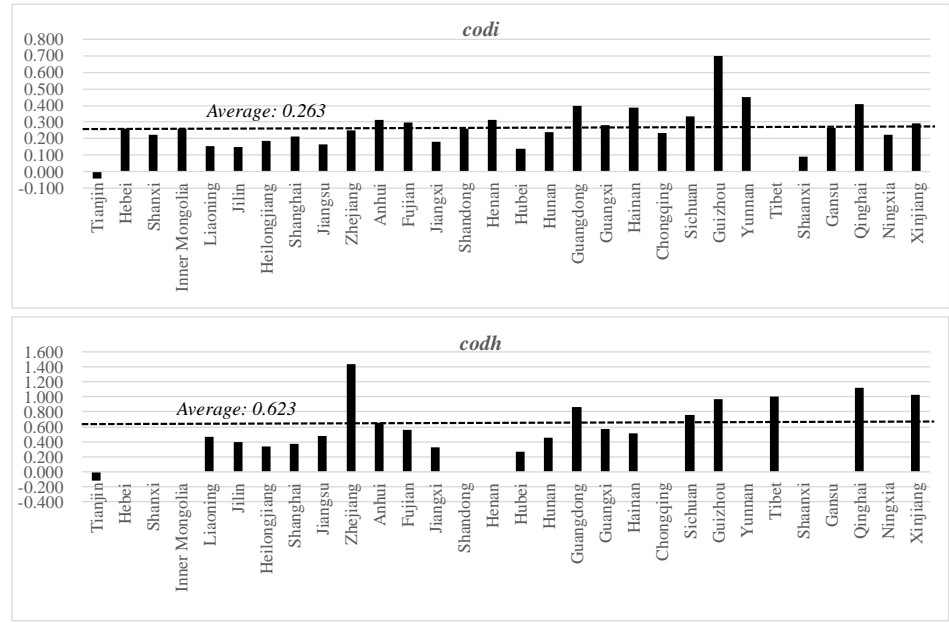

**Figure 3.** *Cont.*

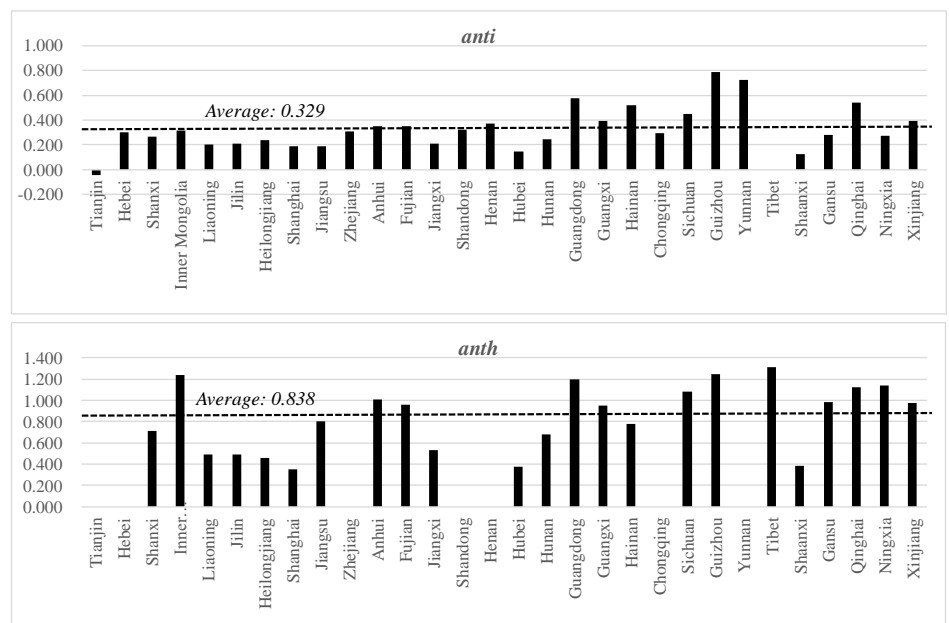

**Figure 3.** Contribution ratios of capacity shortage to water pollution. Note: *codi*: Industrial chemical oxygen demand (COD), tons per million persons; *codh*: Household COD, tons per million persons; *anti*: Industrial ammonia nitrogen, tons per million persons; *anth*: Household ammonia nitrogen, tons per million persons; Sources: Authors' estimation.

## 5. Conclusions

The main findings from the empirical estimations are summarized as follows. First, all EKC estimations with provincial panel data identified the existence of the inverted-U-shaped relationship between water pollution and income with reasonable turning points. Second, the fixed-effect models confirmed that the majority of provinces had more serious water pollution than Beijing as province-specific effects. Third, the alternative models revealed that industrial and household pollution were associated with the industrialization and urbanization degrees, respectively, and, more importantly, both pollutions were significantly affected by the pollution-control capacity. Fourth, the factor analysis demonstrated that the capacity shortage for pollution control is one of the pollution factors that accounted for (a) about 30% of industrial COD and industrial ammonia nitrogen; (b) about 60% of household COD; and (c) about 80% of household ammonia nitrogen.

China still has much policy space and room to mitigate water pollution in terms of COD and ammonia nitrogen, by building the capacity for pollution control through the development and training of human resources. Capacity building contributes to water pollution mitigation through various channels, such as enhancing environmental awareness, developing environmental technologies, and raising regulatory the powers and governance of environmental policies.

The limitations of this study include the shortage of detailed research on individual provinces and regions. China has regional heterogeneity in terms of pollution levels and in the factors affecting them; in addition, there are differences in its policy priority areas, such as in the 3Rs3Ls. Examining the complexity of pollution mechanisms and the policy performances of specific regions through detailed case studies would make it possible to develop firm region-specific recommendations and prescriptions for the management of water pollution in China.

**Author Contributions:** Conceptualization, H.T., T.F., H.A. and J.-J.M.; methodology, H.T., T.F. and H.A.; software, H.T.; validation, H.T., T.F. and H.A.; formal analysis: H.T.; investigation, H.T. and H.A.; resources, J.-J.M.; data curation, J.-J.M.; writing—original draft preparation, H.T.; writing—review and editing, H.T., T.F. and H.A.; supervision, H.T., T.F. and H.A.; project administration, H.T., T.F. and H.A. All authors have read and agreed to the published version of the manuscript.

**Funding:** This research received no external funding.

**Institutional Review Board Statement:** Not applicable.

**Informed Consent Statement:** Not applicable.

**Data Availability Statement:** Not applicable.

**Conflicts of Interest:** The authors declare no conflict of interest.

## Appendix A

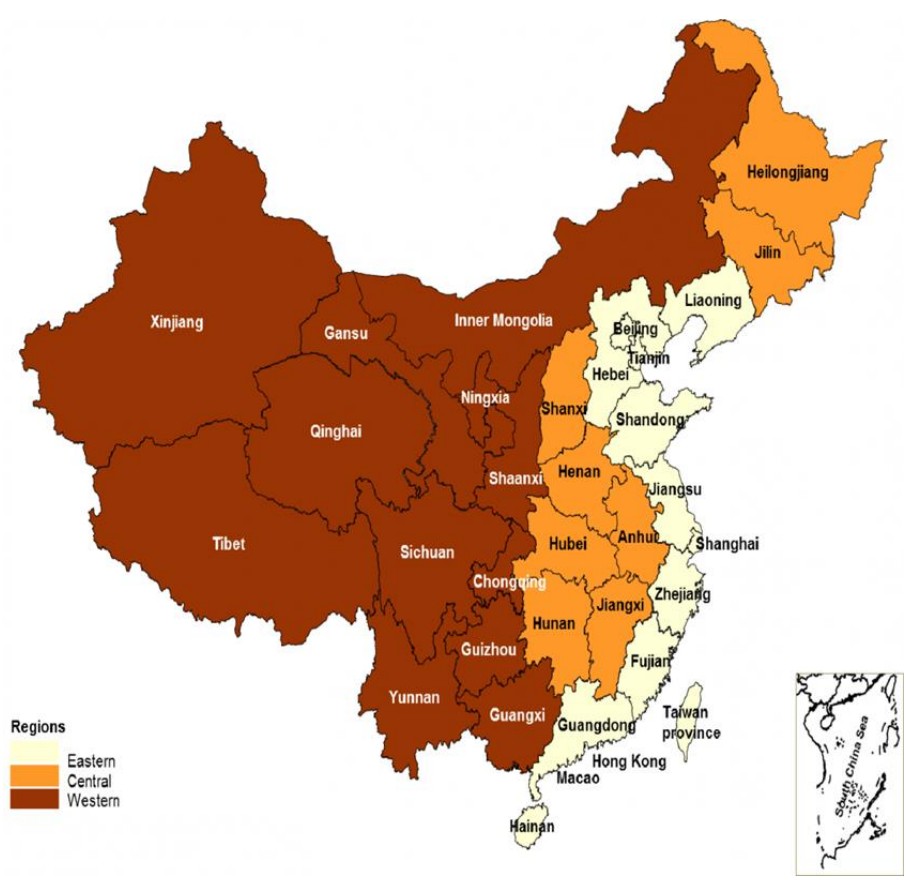

**Figure A1.** Location of Chinese provinces. Sources: China Statistical Yearbook.

## Appendix B

**Table A1.** Estimation on daily treatment capacity of sewage.

| Estimation | *codi* | | *anti* | |
|---|---|---|---|---|
| Methodology | OLS | PPML | OLS | PPML |
| *ypc* | 5.600 ** | 5.631 *** | 3.376 * | 4.311 ** |
| | (2.161) | (3.665) | (1.877) | (2.091) |
| $ypc^2$ | −0.350 ** | −0.351 *** | −0.250 *** | −0.297 *** |
| | (−2.852) | (−4.735) | (−2.842) | (−2.974) |
| *ind* | 0.035 *** | 0.036 *** | 0.048 *** | 0.048 *** |
| | (5.163) | (8.879) | (10.051) | (9.408) |
| *tcs* | 0.825 *** | 0.824 *** | 0.983*** | 1.000 *** |
| | (11.123) | (0.099) | (10.119) | (9.043) |
| Cross-sections | 31 | 31 | 31 | 31 |
| Periods | 2004–2019 | 2004–2019 | 2004–2019 | 2004–2019 |
| Total observations | 487 | 487 | 487 | 487 |

**Table A1.** *Cont.*

| Estimation Methodology | *codh* | | *anth* | |
|---|---|---|---|---|
| | **OLS** | **PPML** | **OLS** | **PPML** |
| *ypc* | 3.646 *** | 3.446 *** | 8.131 *** | 8.168 *** |
| | (8.617) | (3.985) | (8.951) | (6.737) |
| $ypc^2$ | −0.208 *** | −0.201 *** | −0.431 *** | −0.433 *** |
| | (−9.118) | (−4.645) | (−9.916) | (−7.124) |
| *urb* | 0.017 *** | 0.013 *** | 0.018 *** | 0.018 *** |
| | (6.425) | (4.355) | (4.575) | (4.632) |
| *tcs* | 0.134 *** | 0.173 *** | 0.114 * | 0.116 ** |
| | (4.154) | (3.621) | (1.975) | (2.062) |
| Cross-sections | 31 | 31 | 31 | 31 |
| Periods | 2004–2019 | 2004–2019 | 2004–2019 | 2004–2019 |
| Total observations | 479 | 479 | 479 | 479 |

Note: *tcs* denotes the daily treatment capacity of sewage" (cubic meter per million persons, log-term, one-year lagged). ***, **, and * denote statistical significance at 99, 95, and 90% level, respectively. T-statistics are in the parentheses. Sources: Authors' estimation.

## Appendix C

**Table A2.** Estimation on waste water discharge.

| Estimation Methodology | *codi* | | *anti* | |
|---|---|---|---|---|
| | **OLS** | **PPML** | **OLS** | **PPML** |
| *ypc* | −0.656 | −0.032 | 1.437 | 0.894 |
| | (−0.342) | (−0.005) | (0.145) | (0.086) |
| $ypc^2$ | 0.005 | −0.024 | −0.095 | −0.069 |
| | (0.056) | (−0.080) | (−0.204) | (−0.143) |
| *wwdi* | 1.002 *** | 0.993 *** | 1.157 *** | 1.181 *** |
| | (21.573) | (12.244) | (7.968) | (11.729) |
| *edu* | −0.522 | −0.513 *** | −0.453 ** | −0.445 ** |
| | (−1.581) | (−3.668) | (−2.374) | (−2.237) |
| Cross-sections | 31 | 31 | 31 | 31 |
| Periods | 2013–2016 | 2013–2016 | 2013–2016 | 2013–2016 |
| Total observations | 124 | 124 | 124 | 124 |
| **Estimation Methodology** | *codh* | | *anth* | |
| | **OLS** | **PPML** | **OLS** | **PPML** |
| *ypc* | 3.249 * | 2.733 | 3.634 | 3.709 |
| | (2.701) | (0.710) | (1.324) | (0.625) |
| $ypc^2$ | −0.156 * | −0.127 | −0.154 | −0.157 |
| | (−2.528) | (−0.707) | (−1.134) | (−0.564) |
| *wwdh* | 0.330 *** | 0.334 * | 0.297 *** | 0.299 |
| | (8.950) | (1.857) | (10.197) | (1.280) |
| *edu* | −0.291 | −0.442 *** | −0.648 | −0.656 *** |
| | (−1.895) | (−5.005) | (−2.032) | (−5.252) |
| Cross-sections | 31 | 31 | 31 | 31 |
| Periods | 2013–2016 | 2013–2016 | 2013–2016 | 2013–2016 |
| Total observations | 124 | 124 | 124 | 124 |

Note: *wwdi* denotes the industrial discharge of waste water and *wwdh* indicates its household discharge (tons per million persons, log-term, one-year lagged). ***, **, and * denote statistical significance at 99, 95, and 90% level, respectively. T-statistics are in the parentheses. Sources: Authors' estimation.

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
