# Peer review of "Water Pollution and Pollution–Control Capacity in Chinese Provinces: Panel Estimations of Provincial Environmental Kuznets Curves"

_sustainability, doi:10.3390/su15053979_

Round 1

Reviewer 2 Report

The paper describes water pollution and pollution control capacity
in Chinese provinces based on COD and ammonium levels using environmental Kuznets curves.

My specific comments are given below:
- Move Table 1 from the Introduction part to section 3.1. Methodology and Data.
- Add a demonstrative map to show the locations of selected provinces for this study.
- Give more information about the industry types, applied treatment processes for industry and household wastewater discharges (including concentration levels).
- The results of the presented study should be compared and discussed with the previous similar studies as shown in Table 2 of the paper.
- The pollution control capacity is represented by the number of graduates of higher education. A better parameter, such as treatment plant capacities and efficiency, could be selected. This assumption needs to be validated.
- Additionally, industrialization and urbanization factors were described indirectly in the model. Direct measurements of industrial and household wastewater flow rates could be used. This also needs verification.
- What is the significance of Figure 1? Why does the lag pattern differ for COD and ammonium with respect to the same number of higher education graduates?
- Please add a graph to show the existence of the inverted-U shaped relationships between water pollution and income levels.

- Improve discussion and conclusion sections.

-

Reviewer 3 Report

Dear authors,

Could you please make your article more readable, more structured, and easy to follow? 

It will be helpful to make sentences in the article straight, without complex chains. Please try to avoid the "respectively" referencing, which requires effort to understand connections.   

For example, this sentence:

"The capacity shortage for pollution control, as a pollution factor, accounted for 30% of industrial COD and ammonia nitrogen and for 60% and 80% of household COD and ammonia nitrogen, respectively." 

Could be presented, if I understand correctly these connections, in this way: 

The pollution factors, which represent the capacity shortage for pollution control, are accounted as a) 30% of industrial COD and ammonia nitrogen; b) 60% of household COD;  c) 80% of ammonia nitrogen. 

Please review similar sentences and make them easy to follow. 

You mentioned in the Introduction that " We performed the following steps: (1) the EKC was estimated econometrically from the provincial panel data using a fixed-effect model; (2) the province-specific pollution effect was extracted from the fixed effect, which is not affected by the provincial income level, on the EKC; (3) the alternative EKC was re-estimated by replacing the fixed-effects with the possible contributors to the province-specific pollution, such as the capacity for pollution control, industrialization degree (for industrial discharges), and urbanization degree (for household discharges); and (4) the contribution of capacity shortage for pollution control to the province-specific pollution level was quantified through factor analysis. "

Where are all these steps in the manuscript?

Please make the related sub-headings with the steps, which you mentioned. The structured proper explanations should be within each sub-section, what you did. It will help readers to understand better the logical structure of your work, and to follow your work activities. 

The sub-headings could be something as: 

(1) The fixed-effect model for EKC estimation;

(2) The province-specific pollution effect extraction

(3) The alternative EKC re-estimation

(4) The pollution factors analysis

Moreover, please use more visual graphics, and box plots,  instead of tables. You may share your tables with data as attachments, and support files. Within each Figure (graphics, and box plots) please give the proper explanations, of what readers see in the figures. 

These types of research works are very important for governmental agencies, politicians, administrators, and decision-makers. They have to understand properly this scientific research for further practical implementation activities. You will have more readers if you make your article more user-friendly. The well-structured manuscript with good figures will be easy to follow. 

Good luck !

Round 2

Reviewer 2 Report

My review comments were fully taken into consideration in the revised form.

Please correct the term "pollutions" in the whole manuscript as "pollution".

Author Response

We express greatest thanks for your variavle comments.

We corrected the term "pollutions" in the whole manuscript as "pollution".

Thanks again.

Reviewer 3 Report

Dear Authors,

your article was substantially improved, it would be interesting to read by many researchers. 

Good luck!

Author Response

We express the greatest thanks for all of your commnets, and the manuscript has been highly improved.

Thanks again.